# Accelerating Effects of Flow Behavior Index *n* on Breakup Dynamics for Droplet Evolution in Non-Newtonian Fluids

**DOI:** 10.3390/ma15134392

**Published:** 2022-06-21

**Authors:** Jinsong Zhang, Yufeng Han, Zhiliang Wang

**Affiliations:** 1School of Mechatronic Engineering and Automation, Shanghai University, Shanghai 200444, China; zhangjs@shu.edu.cn (J.Z.); yufeng@shu.edu.cn (Y.H.); 2School of Mechanics and Engineering Science, Shanghai University, Shanghai 200444, China

**Keywords:** non-Newtonian fluid, flow behavior index, micro-droplet, breakup dynamics

## Abstract

This paper studied the evolution of NaAlg solution micro-droplet in a coaxial microchannel. The Bird–Carreau model was used to characterize the flow properties of NaAlg solution. As the mass fraction decreased, the flow behavior index *n* also decreased, indicating that the NaAlg solution was increasingly shear-thinning. There were three stages during the micro-droplet evolution, which were the growth stage, the squeezing stage, and the pinch-off stage. This paper led the flow behavior index *n* to estimate the effects of rheological property on the breakup dynamics of micro-droplet. We proposed two new prediction models of the minimum neck width *w_m_* which were affected by |*n*| in the squeezing and pinch-off stages for the non-Newtonian fluids. In addition, this paper indicated the rate ratio *Q_d_*/*Q_c_* was another factor on the *w_m_* model in the squeezing stage and the *H*(*λ*) of Stokes mechanism was a function governed by |*n*|^2^ in the pinch-off stage.

## 1. Introduction

In the microfluidic technology, it was one of the hot issues to study the flow behavior of two-phase flow in microchannels, especially, the breakup process of micro-droplet. On the one hand, the micro-droplet is widely used in chemical reaction, pharmaceutical technology, ink-jet technology, etc. On the other hand, the micro-droplet breakup behavior can reflect abundant physical mechanisms [1,2].

Up to now, the breakup dynamics of the micro-droplet in Newtonian fluids has been well documented and four universal dynamical regimes are obtained [3,4,5,6]. (1) Potential Flow mechanism, which was controlled by the inertial force and expressed by a power-law relationship between minimum neck width and remaining time with an exponent of 2/3. (2) Viscid Thread mechanism, which was dominated by the viscous force and expressed by a linear relationship between minimum neck width and remaining time. (3) Inertial Viscous mechanism, which was governed by the coupling effects among the capillary force, the inertia force and the viscous force. The minimum width of the micro-droplet neck is linearly dependent on the remaining time. (4) Stokes mechanism, which was guided by the coupling effects between the viscous force and the interfacial tension, and expressed by the minimum neck width depends on the remaining time linearly. 

In recent years, the research of non-Newtonian fluids on the breakup dynamics in micro-droplets was booming to match the development of chip lab, drug delivery, and high-speed inkjet printing. The non-Newtonian fluids usually presented some complex properties, such as the shear thickening, the viscoelasticity, and the shear thinning, to affect the breakup behavior of micro-droplet and change the micro-droplet size and mono-dispersion. The related research in non-Newtonian fluids was focused on the rheological properties, e.g., the viscoelasticity and the shear thinning. When a viscoelastic fluid acts as the dispersed phase, it can affect the neck breakup dynamics of micro-droplets [7]. In the squeezing stage, the relationship between minimum neck width and remaining time obeyed an exponent law [8,9,10]. In the pinch-off stage, the viscous, inertial, and elastic forces resisted the interfacial tension to hinder the breakup of micro-droplet neck [11].

As the shear-thinning fluid was the dispersed phase, the breakup behavior of it was similar to those of the Newtonian fluid [12,13,14,15], but their difference were the breakup time, micro-droplet size, and satellite micro-droplet state [16]. The viscosity of shear-thinning fluid varies with shear rate, thus affecting the micro-droplet breakup. There were the dissimilar models between the minimum neck width and the remaining time in different stages of micro-droplet formation. In the squeezing stage, the typical power-law relationship between the minimum neck width and the remaining time was found, and its power exponent increased with the increasing of the mass fraction and the capillary number in continuous phase [17]. In the pinch-off stage, the shear thinning raises the speed of micro-droplet breakup and reduced the size of micro-droplet by the minimum neck width correlated to the remaining time linearly [15].

As mentioned above, there are a few articles studying the breakup dynamics of micro-droplet in the shear-thinning non-Newtonian fluids. Moreover, the rheological property of shear-thinning fluids was described by the power-law model in them [17,18]. This paper studied the rheological property of non-Newtonian fluid by the Bird–Carreau model in the vertical and coaxial microchannel, and discussed the accelerated effects of flow behavior index *n* on the breakup dynamics of micro-droplet in different evolution stages.

## 2. Experiments and Simulation

The experiment devices of two-phase flow in microchannel consisted of the fluid driving unit, the microchannel sample, and the observation unit (Figure 1). 

The microchannel sample was vertical and coaxial [19,20], in which the dispersed phase flowed in the inner tube with the NaAlg solution and the continuous phase flowed in the outer tube with Oil. The inner diameters of the outer tube and the inner tube, respectively, were 1 mm and 0.42 mm. The driving pumps were the syringe pump (LSP02-2A, GELAN, Shanghai, China) and the advection pump (SP-6015, SANPTAC, Shanghai, China). There were four group experiments with the different mass fraction of NaAlg solution (0.1 wt%, 0.5 wt%, 1.0 wt% and 1.5 wt%). The experiment parameters were in the range of *Q_d_* = 2 mL/h for the dispersed phase and *Q_c_* = 10~100 mL/h for the continuous phase. 

A high-speed camera (Phantom V611-16G-M, AMETEK, USA) is used, with a micro-lens (AT-X M100 PRO-D, Tokina, Tokyo, Japan), to capture images of the two-phase flow. All data are real-time stored in the computer. The two-phase interfacial tension coefficient *σ* is measured using the pendant drop method. The instrument is an interfacial tension meter (SL200KS, KIM, Boston, MA, USA). The NaAlg solution dynamic viscosity *η* is measured by a rotational rheometer (AR-1500ex, AT, Newcastle, USA). The viscosity of the continuous phase Oil is measured by rotational viscometer (NDJ-9S, GengGeng, Shanghai, China). The physical properties of two-phase flow are depicted in Table 1.

Figure 2 plotted the relationship between the viscosity *η* and the shear rate *γ* in the NaAlg solution with the different mass fractions. As the shear rate *γ* gradually increased, the viscosity *η* of the NaAlg solution decreased rapidly and approached a constant. Aqueous solutions of NaAlg with different mass fractions were used as the dispersed phase, which were demonstrated as shear-thinning non-Newtonian fluids.

The commonly used models to describe the flow behavior of shear-thinning fluids were the power–law [21,22] or Ostwald de Waele model [23,24], Bird–Carreau, Herschel–Bulkley models, etc. In Figure 2, the experimental data agree well with the predicted values of the Bird–Carreau model, and the relationship between the fluid viscosity *η* and the shear rate *γ* could be expressed as [25,26,27]:(1)$$\eta ={\left(1+{\left(\lambda_{r}\gamma \right)}^{2}\right)}^{\frac{n-1}{2}}\left(\eta_{0}-\eta_{\infty }\right)+\eta_{\infty }$$
where *n* is the flow behavior index, *λ_r_* is the relaxation time of material (s), *η*_0_ is the zero-shear viscosity (Pa⋅s), and *η_∞_* is the infinite shear viscosity (Pa⋅s).

The flow behavior index *n* was an important parameter to describe the rheological property of fluid. In general, as *n* = 1, the fluid was defined as the Newtonian fluid. As *n* < 1, the fluid was defined as the shear-thinning non-Newtonian fluid. The smaller the *n* value, the faster the viscosity of the fluid decreased with increasing shear rate. The fitting parameters of the Bird–Carreau model were listed in Table 1. In Table 1, as the mass fraction decreased, the flow behavior index *n* also decreased, indicating that the NaAlg solution was increasingly shear-thinning. There were two positive values of *n* and two negative values of *n*. This was the first paper to report *n* had more values below zero. 

The finite element analysis based Comsol software is used to simulate the present microchannel flow. A two-dimensional axisymmetric model is drawn according to the experimental microchannel structure parameters (Figure 3).

The Phase field model is adopted to capture the liquid–liquid interface. The governing equations of the Phase field model on the two-phase flow are given as follows:

Equation of continuity:(2)∇⦁u=0

Equation of motion:(3)$$\rho \frac{\partial u}{\partial t}+\rho \left(u⦁\nabla \right)u=\nabla ⦁\left[-pI+\eta \left(\nabla u+{\left(\nabla u\right)}^{T}\right)\right]+F_{st}$$
where *ρ* is the density(kg/m^3^), *u* is the velocity of liquid (m/s), *p* is the pressure (Pa), *I* is the unit vector, *η* is the viscosity (Pa⋅s), and *F_st_* is the Interfacial tension (N/m^3^). *F_st_* is the surface tension force approximated as a body force in the vicinity of the interface.

Equation of density *ρ* and viscosity *η*:(4)$$\left\{ \begin{array}{l} \rho =\rho_{d}+(\rho_{c}-\rho_{d})\varphi \\ \eta =\eta_{d}+\left(\eta_{c}-\eta_{d}\right)\varphi \end{array}\right.$$
where *φ*—represents the phase field variables, with *φ* = 1 for the continuous phase, *φ* = −1 for the dispersed phase, and 0 < *φ* < 1 for the liquid–liquid interface.

## 3. Results and Discussion

### 3.1. Formation and Breakup Mechanism of Micro-Droplet

The NaAlg micro-droplets had been found in all four group experiments under the same experimental and simulation conditions. The breakup moment of previous micro-droplet was set to be the starting point (*t* = 0 s) for new micro-droplet formation, and the breakup moment of new micro-droplet was set to be the terminal point (*t* = *T* s). Thus, one period time *T* could be recorded to describe the formation and breakup for the micro-droplet. 

Figure 4 was the diagram of one period of micro-droplet formation. The breakup process could be characterized by the initial neck width *w*_0_, the minimum neck width *w_m_* and the remaining time *T* − *t*. The initial neck width *w*_0_ and the minimum neck width *w_m_* were obtained by averaging three measurements by ImageJ software.

The numerical simulation had been introduced to restructure the formation and breakup for the micro-droplet. The result images of experiments and numerical simulation with the different mass fractions of NaAlg solution were compared in Figure 5. It was that the period time of micro-droplet formation both in the experiments and numerical simulation were very similar, but the shapes of micro-droplet in them were slightly different. The comparison revealed there were the same rules for the breakup dynamics of micro-droplet with the different experiment parameters.

In order to further understand the breakup mechanism of shear-thinning micro-droplets in the microchannel, the breakup process of the micro-droplet neck was next quantitatively analyzed. Figure 6 plotted the relationship between the minimum neck width *w_m_* and the remaining time *T* − *t* within one period time *T* by the data processing from Figure 5. The formation and breakup of NaAlg micro-droplets in one period could be separated for three stages [22,28,29,30], which were the growth stage(*T* − *t* > 270 ms), the squeezing stage(20 ms < *T* − *t* < 270 ms), and the pinch-off stage(*T* − *t* < 20 ms). 

In Figure 6, the data in growth, squeezing, and pinch-off stages were signed by the colors of blue, red, and black, respectively. It was true that the minimum neck width *w_m_* in different evolution stage obeyed the dissimilar evolution mechanism. In the growth stage, the minimum neck width of the neck was linear with remaining time: *w_m_* = *k*_1_ (*T* − *t*) + *b*. In the squeezing stage, the minimum width was power–law dependent with the remaining time: *w_m_* = *k*_2_(*T* − *t*)^α^. In the pinch-off stage, the minimum neck width transitioned from a power–law relationship to another linear relationship with remaining time: *w_m_* = *k*_3_ (*T* − *t*). This pattern was observed for NaAlg solutions with different mass fractions and different flow ratios (*Q_d_*/*Q_c_*).

In the growth stage, the head of micro-droplet expanded downstream slowly and then formed a shallow neck gradually. The growth stage occupied a much longer time for micro-droplet formation compared with other two stages during one period. The minimum neck width *w_m_* had a linear relation to the remaining time *T* − *t* (Equation (5)). With the remaining time *T* − *t* increasing, the minimum neck width *w_m_* increases. This law also had been found in other researches on both the Newtonian fluids [31,32] and non-Newtonian fluids [21].
(5)$$w_{m}=k_{1}\left(T-t\right)+b$$

Subsequently, in the squeezing stage, the continuous phase pushed the head of micro-droplet downward along the microchannel to shrink being a visible neck and decrease the minimum neck width *w_m_*. Finally, in the pinch-off stage, the viscous force retarded the breakup of micro-droplet, and the interfacial tension produced the minimum neck width *w_m_* declining to zero rapidly. The minimum neck width *w_m_* was proportional to the remaining time *T* − *t*. 

However, the effects of behavior index *n* on the minimum neck width *w_m_* in the squeezing and pinch-off stages were not clear. This paper was focusing on the different laws of micro-droplets in the squeezing and pinch-off stages detailly.

### 3.2. Squeezing Stage Dynamics

In the squeezing stage, the swelled head of micro-droplet reduced the gap between the micro-droplet and the microchannel wall and retarded the continuous phase flowing smoothly. The pressure difference of continuous phase between the rear of micro-droplet and the head of micro-droplet increased rapidly; as a result, the micro-droplet neck was squeezed to becomes thinner (Figure 7). 

The morphology of micro-droplet would be changed by the competition of driving force and resisting force. The driving force by continuous phase fluid included the squeezing force (∆*P*), the shear stress (*τ*_a_), and the tensile stress (*τ*_r_). On the contrary, the resisting force was the viscous force of dispersed phase which retard the neck of micro-droplet changing. Here, the squeezing force ∆*P*, the shear stress *τ*_a_, and the tensile stress *τ*_r_ can be expressed [21,33,34,35].
(6)$$\Delta P\approx \eta_{c}Q_{c}L/(h\varepsilon^{3})$$
(7)$$\tau_{a}=\frac{u_{c}Q_{c}}{2\varepsilon [w_{c}{}^{2}-0.785{\left(w_{c}-2\varepsilon \right)}^{2}]}$$
(8)$$\tau_{r}=3\eta_{c}Q_{c}/w_{c}{}^{3}$$
where *h* is the depth of the channel (m), *L* is the length of liquid film (m), *η*_c_ is the viscosity of continuous phase (Pa⋅s), and *ε* is the thickness of liquid film (m). 

Importing data of Figure 7 for calculation, the squeezing force ∆*P* was above 73.8 Pa, the shear stress *τ*_a_ was in the range of 0~8.5 Pa, and the tensile stress *τ*_r_ was in the range of 0.56–2.3 Pa. In the three driving forces, the squeezing force ∆*P* was the main driving force compared with the shear stress *τ*_a_ and the tensile stress *τ*_r_. 

The following is a quantitative analysis of the squeezing stage of micro-droplet breakup. Regarding to the different Newtonian and non-Newtonian fluids, the prediction model of minimum neck width *w_m_* in the squeezing stage has been proposed as the same one [36,37].
(9)$$w_{m}=k{\left(T-t\right)}^{\alpha }$$

As is well known, the flow behavior index *n* was a dimensionless parameter to characterize the rheological property of fluid, this paper led the flow behavior index *n* into the micro-droplet breakup dynamics. There were four mass fractions of NaAlg solution corresponding to the different flow behavior indices *n* (1.5 wt% with *n*_1_ = 0.437, 1.0 wt% with *n*_2_ = 0.162, 0.5 wt% with *n*_3_ = −0.567, 0.1 wt% with *n*_4_ = −0.583). Figure 8 showed the relationships between the minimum neck width *w_m_* and the remaining time *T* − *t* in the squeezing stage with the four flow behavior indices *n* under different experiment parameters. In the same absolute value of flow behavior index |*n*|, the minimum neck width *w_m_* increased with the rate ratio *Q_d_*/*Q_c_* decreasing. Furthermore, the remaining time *T* − *t* increased with the rate ratio *Q_d_*/*Q_c_* increasing. The different exponent *α* in Equation (9) had been fitted by the rate ratio *Q_d_*/*Q_c_* and the remaining time *T* − *t* under the different conditions of |*n*|.

In the same condition of |*n*|, the independent parameters affecting the minimum neck width *w_m_* were not only the remaining time *T* − *t*, but also the rate ratio *Q_d_*/*Q_c_*. In the different conditions of |*n*|, the exponent *α* increased as |*n*| increasing. Overall consideration *Q_d_*/*Q_c_* and |*n*| effects on *w_m_*, the new prediction model had been proposed in the form of Equation (9) by fitting used the multivariate nonlinear regression. Equation (10) stated that *Q_d_*/*Q_c_* and *T* − *t* had the independent effects on the *w_m_* model, |*n*| had the exponent effects on the *w_m_* model, respectively. Compared with Equation (9), Equation (10) added two extra variables which were the experiment parameters *Q_d_*/*Q_c_* and the rheological parameter |*n*|.
(10)$$w_{m}=k{\left(\left(T-t\right)\cdot \frac{Q_{d}}{Q_{c}}\right)}^{\alpha \left|n\right|}$$

It also could be found that the flow behavior index *n* influenced the minimum neck width *w_m_* by rather its value size |*n*| than its value sign +/−. Since the flow behavior index *n* changed from positive to negative accompany with mass fractions decreasing, the rheological property in the non-Newtonian fluid could be characterized by the size of flow behavior index *n*. Hence, the absolute value |*n*| should be introduced to characterize the effects of rheological property exactly on the minimum neck width *w_m_* in the squeezing stage.

The new prediction model and its accuracy were discussed in Figure 9. In it, the horizontal axis was the model prediction values, and the vertical axis was the experimental values. In comparison, the model prediction values matched the experimental values well with an error less than |15%|. 

Our new prediction model of the minimum neck width *w_m_* in equation 10 could be applied in both the Newtonian and non-Newtonian fluids. Table 2 listed the different *w_m_* model with different experiment conditions in the Newtonian and non-Newtonian fluids. For *n* = 1 and regardless to the effects of experiment parameters *Q_d_*/*Q_c_*, Equation (10) could be simplified to Equation (9), which was used in the Newtonian fluid [17,38,39]. For 0 < *n* = constant < 1 and not introducing experiment parameters *Q_d_*/*Q_c_*, Equation (10) could be simplified to Equation (9), which was in non-Newtonian fluid [21,37]. Therefore, the normalization model of *w_m_* could be widely used for the micro-droplet breakup dynamics in the squeezing stage. 

### 3.3. Pinch-Off Stage Dynamics

With the time passing by, the breakup of micro-droplet entered the pinch-off stage, in which the minimum neck width *w_m_* of micro-droplet would approach zero. At the beginning of pinch-off stage, the front neck and the rear neck became thinner in sequence (Figure 10). Closing to the range of neck breakup, the micro-droplet was subjected to the combined action of the viscous force of the dispersed phase and the interfacial tension of the two phases. This phenomenon agreed with the Stokes mechanism of micro-droplet breakup in the Newtonian fluid [40]. First, the effect of viscous forces was more pronounced, slowing down the breakup process of micro-droplet. Then, the effect of interfacial tension was more pronounced, rapidly reducing the minimum neck width and causing the breakup of micro-droplet.

The following is a quantitative analysis of the pinch-off stage of micro-droplet breakup. As shown in Figure 11, the minimum neck width *w_m_* of micro-droplet was proportional to the remaining time *T* − *t*. The minimum neck width *w_m_* of micro-droplet increased to respond the |*n*| increasing at the conditions of *Q_d_*/*Q_c_* not changing.

Processing the data of Figure 11 with different *Q_d_*/*Q_c_* and |*n*| values, the relationship between the minimum neck width *w_m_* and the remaining time *T* − *t* were listed in Table 3. It was interesting that *w_m_* model was not change under the same *n* values, regardless of *Q_d_*/*Q_c_* changing. The slope of linear equation increased with different *|n|* increasing as *Q_d_*/*Q_c_* was constant. As a result, it was rather *|n|* than *Q_d_*/*Q_c_* to bring the significant effects on *w_m_* in the pinch-off stage.

The scaling law of Stokes mechanism in the pinch-off stage was in the following expression [41].
(11)$$w_{m}=H\left(\lambda \right)\frac{\sigma }{\eta_{d}}\left(T-t\right)$$
where *H*(*λ*) is the scaling law coefficient.

For the previous research, *H*(*λ*) was a certain value in the Newtonian fluid related to the two-phase viscosity ratio *λ* [42], or in the non-Newtonian fluid related to the mass fraction *wt* [21,22]. The equations in Table 3 with different *|n|* also proved *H*(*λ*) being a certain value. Therefore, fitting all data with different *|n|* and *Q_d_*/*Q_c_* by using the multivariate nonlinear regression, the new model was conducted to describe the minimum neck width *w_m_* and the remaining time *T* − *t*.
(12)$$\left\{ \begin{array}{l} w_{m}=H\left(\lambda \right)\frac{\sigma }{\eta_{d}}\left(T-t\right)\\ H\left(\lambda \right)=k{\left|n\right|}^{2}+b \end{array}\right.$$
where *k* is the coefficient and *b* is the constant.

According to Equation (12), *H(λ)* was not a constant defined by the viscosity ratio *λ* or the mass fraction *wt*, but was a function positively related to the absolute value square of *n* (|*n*|^2^). The *H(λ)* function with |*n*|^2^ had a highly significant effects on the prediction model of minimum neck width *w_m_*. 

Our new model of *w_m_* and its accuracy in the pinch-off stage were shown in Figure 12. In it, the horizontal axis was the prediction values, and the vertical axis was the experimental values. It was seen that the prediction model of minimum neck width *w_m_* had a high accuracy with an error less than |10%|. 

Since our new model introduced the rheological property parameter |*n*|, Equation (12) became the normalization model, which could be used in both the Newtonian and non-Newtonian fluids. For *n* = 1, the *H*(*λ*) function in Equation (12) collapsed to be a coefficient which was the Stokes scaling law in the Newtonian fluid (Equation (11)). Table 4 listed and processed the literature data by our normalization equation with different rheological models in the non-Newtonian fluid. Figure 13 plotted all data from different literature together by the same *H*(*λ*) function. The data were marked by circle and triangle being the Power rheological model, by star being the Herschel–Bulkley rheological model and by square being the Bird–Carreau rheological model. As shown in Figure 13, it was true that *H*(*λ*) function was positively related to the |*n*|^2^ and was not related to the rheological model. And the k and b values of *H*(*λ*) function from different literature were similar. This proved that our new model with *H*(*λ*) function was available to predict the minimum neck width *w_m_* at different rheological model.

## 4. Conclusions

This paper focused on the shear-thinning non-Newtonian fluid to study the breakup dynamics of NaAlg micro-droplet in a coaxial microchannel, including the growth stage, the squeezing stage and the pinch-off stage. Some conclusions were drawn, as below. 

(1)It was found that as the mass fraction of NaAlg solution decreased, the *n* value changed from 1 > *n* > 0 to 0 > *n* > −1. Moreover, the smaller the *n* value was, the more significant the shear-thinning of fluid was;(2)In the growth stage, the minimum neck width *w_m_* of micro-droplet was linearly positively correlated to the remaining time *T* − *t*, and the influence of flow behavior index *n* was not significant;(3)In the squeezing stage, the new prediction model of minimum neck width *w_m_* was established with an error less than |15%| and it was suit for both the Newtonian and non-Newtonian fluids;
$$w_{m}=k{\left(\left(T-t\right)\cdot \frac{Q_{d}}{Q_{c}}\right)}^{\alpha \left|n\right|}$$(4)In the pinch-off stage, another new prediction model of minimum neck width *w_m_* by *H*(*λ*) function was established with an error less than |10%|. It matched Stokes scaling law and could be used in both the Newtonian and non-Newtonian fluids regardless the rheological models.
$$H\left(\lambda \right)=k{\left|n\right|}^{2}+b$$

## Figures and Tables

**Figure 1 materials-15-04392-f001:**
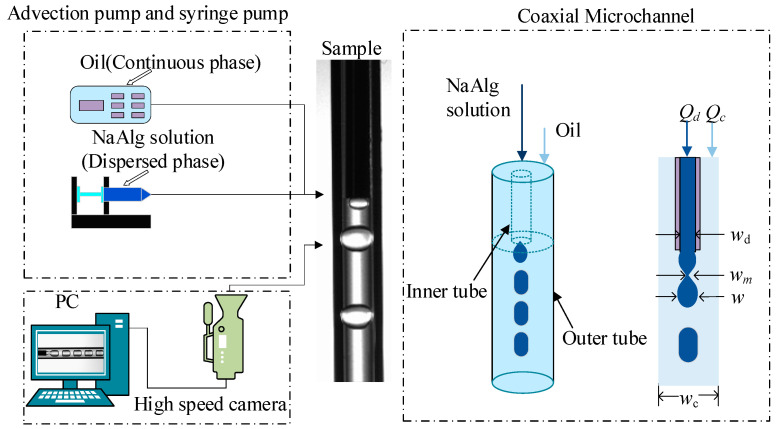
The schematic diagram of experiment devices.

**Figure 2 materials-15-04392-f002:**
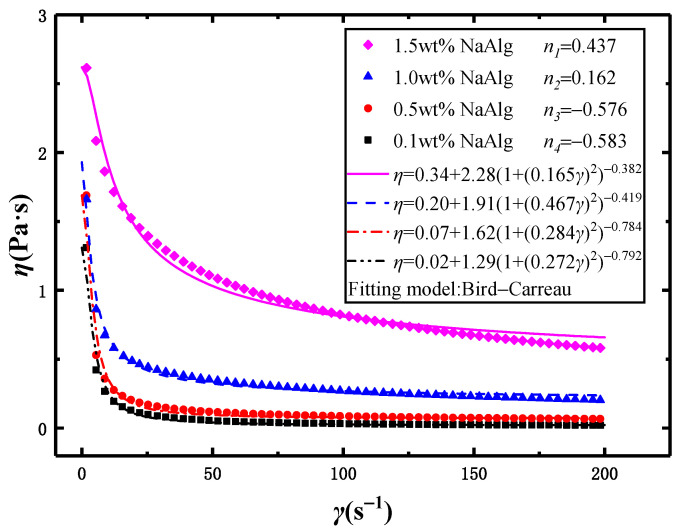
The relationship between the viscosity (*η*) and the shear rate (*γ*) with four mass fractions of the NaAlg solution (The solid line displays the fitting result of the Bird–Carreau model).

**Figure 3 materials-15-04392-f003:**
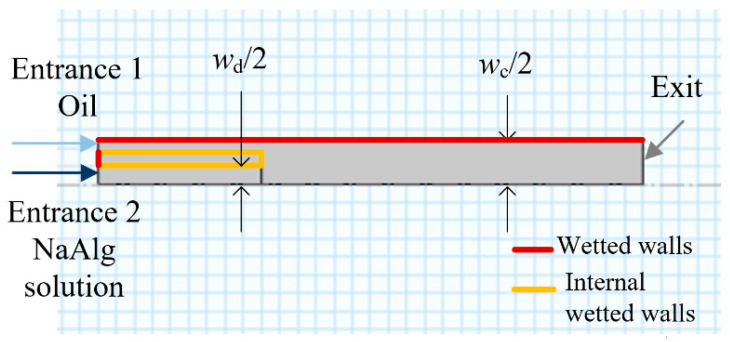
Two-dimensional axisymmetric geometric structure of the microchannel.

**Figure 4 materials-15-04392-f004:**
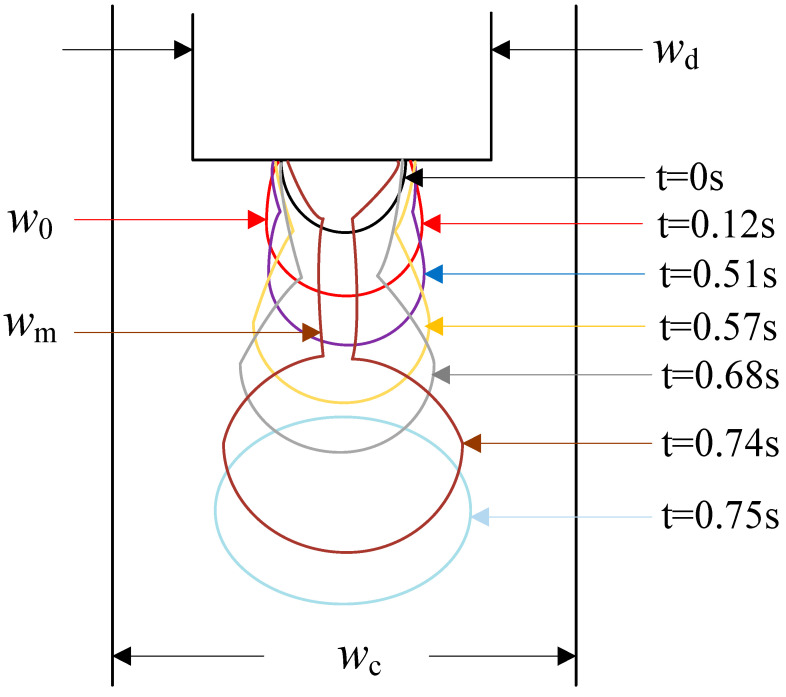
The schematic diagram of NaAlg micro-droplet in the breakup process. (1.5 wt%, *Q_d_* = 2 mL/h, *Q_c_* = 20 mL/h).

**Figure 5 materials-15-04392-f005:**
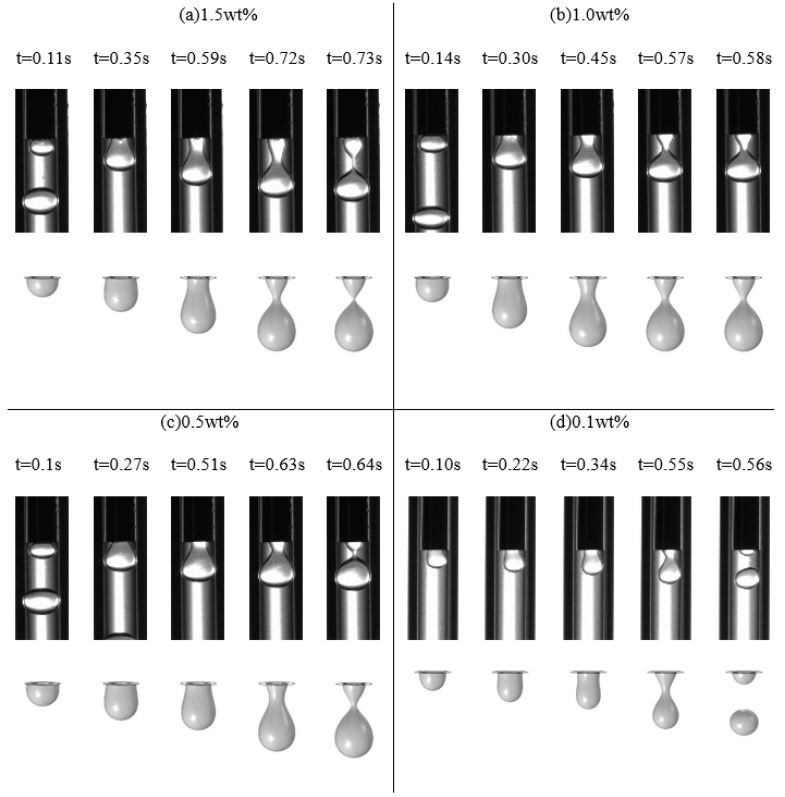
The comparison between the experiment results and numerical simulation results (*Q_d_* = 2 mL/h, *Q_c_* = 20 mL/h). (**a**) 1.5 wt%, *n*_1_ = 0.437 (**b**) 1.0 wt%, *n*_2_ = 0.162 (**c**) 0.5 wt%, *n*_3_ = −0.567 and (**d**) 0.1 wt%, *n*_4_ = −0.583.

**Figure 6 materials-15-04392-f006:**
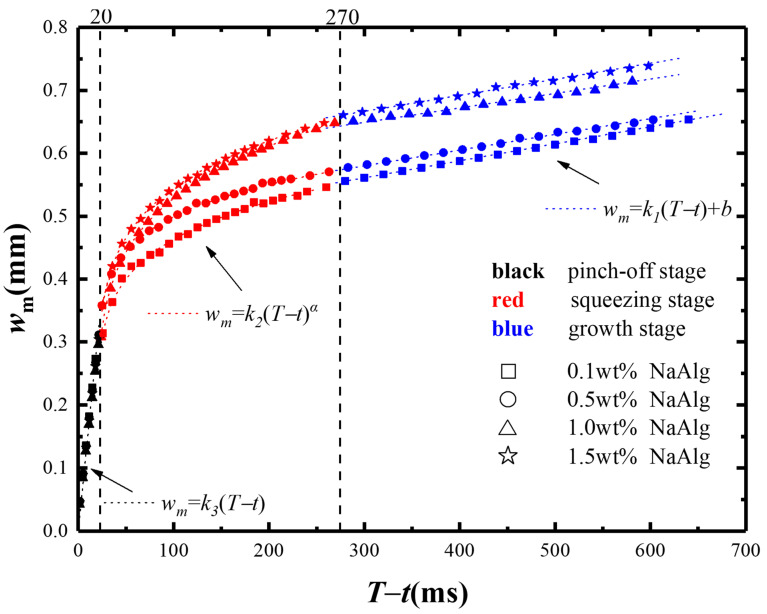
The minimum neck widths *w_m_* versus the remaining time *T* − *t* with four different mass fractions of NaAlg solution (*Q_d_* = 2 mL/h, *Q_c_* = 20 mL/h).

**Figure 7 materials-15-04392-f007:**

The NaAlg micro-droplet evolution in the squeezing stage (*n*_2_ = 0.162, *Q_d_* = 2 mL/h, *Qc* = 40 mL/h).

**Figure 8 materials-15-04392-f008:**
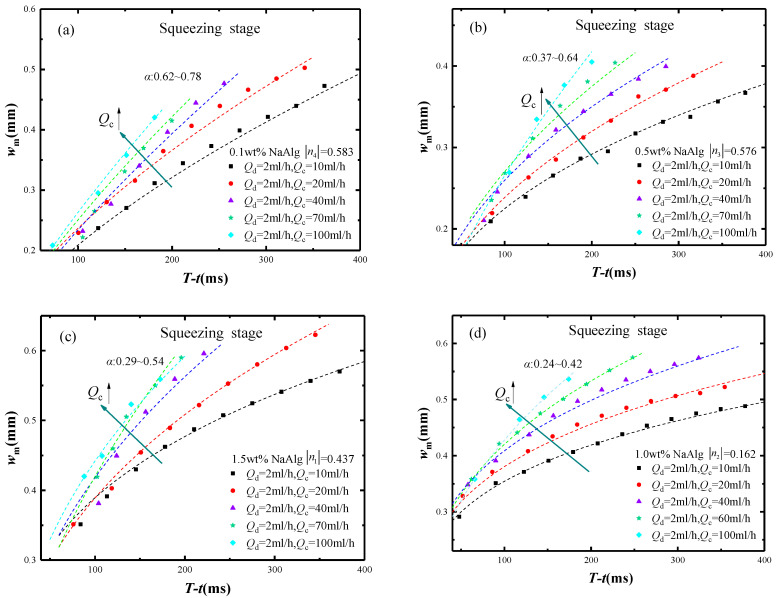
The relationship between wm and *T* − *t* with different *Q_d_*/*Q_c_* in the squeezing stage, (**a**) |*n*_4_| = 0.583, (**b**) |*n*_3_| = 0.576, (**c**) |*n*_1_| = 0.437 and (**d**) |*n*_2_| = 0.162.

**Figure 9 materials-15-04392-f009:**
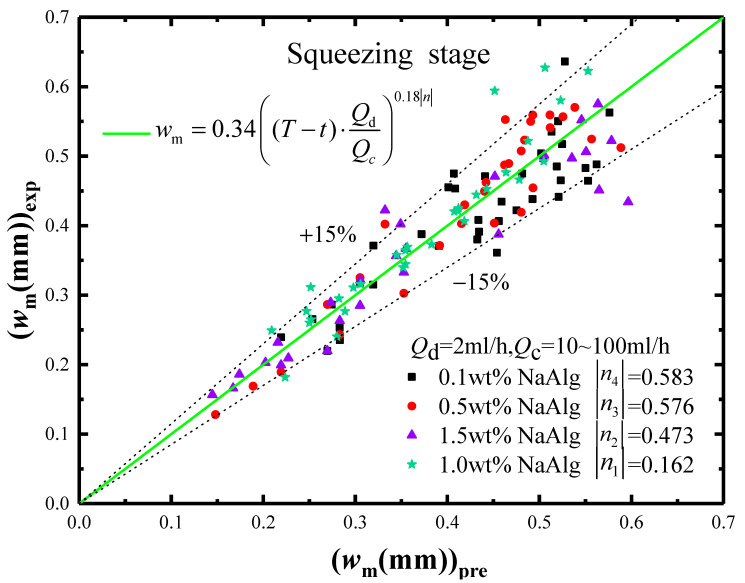
The accuracy analysis of *w_m_* prediction model in the squeezing stage.

**Figure 10 materials-15-04392-f010:**

The NaAlg micro-droplet evolution in the pinch-off stage (*n*_2_ = 0.162, *Q_d_* = 2 mL/h, *Q_c_* = 40 mL/h).

**Figure 11 materials-15-04392-f011:**
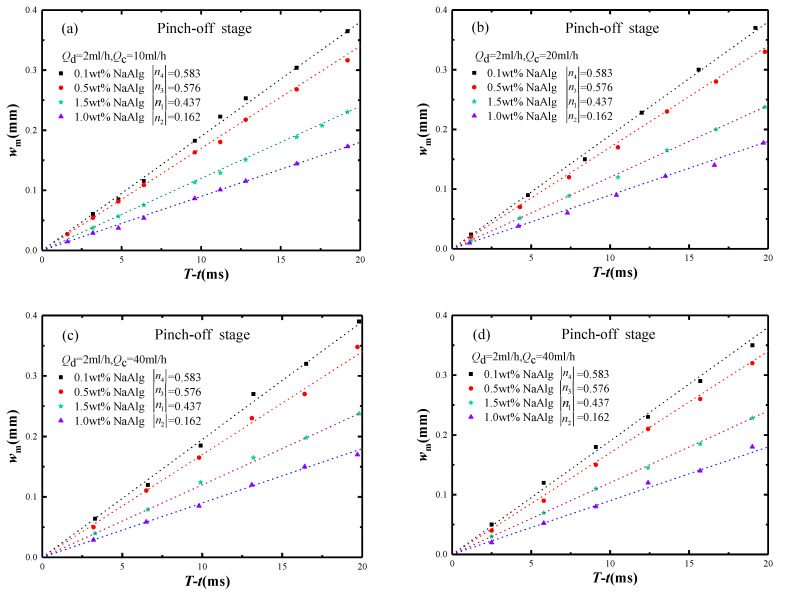
The relationship between *w_m_* and *T* − *t* with different |*n*| in the pinch-off stage, (**a**) *Q_d_* = 2 mL/h, *Q_c_* = 10 mL/h, (**b**) *Q_d_* = 2 mL/h, *Q_c_* = 20 mL/h, (**c**) *Q_d_* = 2 mL/h, *Q_c_* = 40 mL/h and (**d**) *Q_d_* = 2 mL/h, *Q_c_* = 100 mL/h.

**Figure 12 materials-15-04392-f012:**
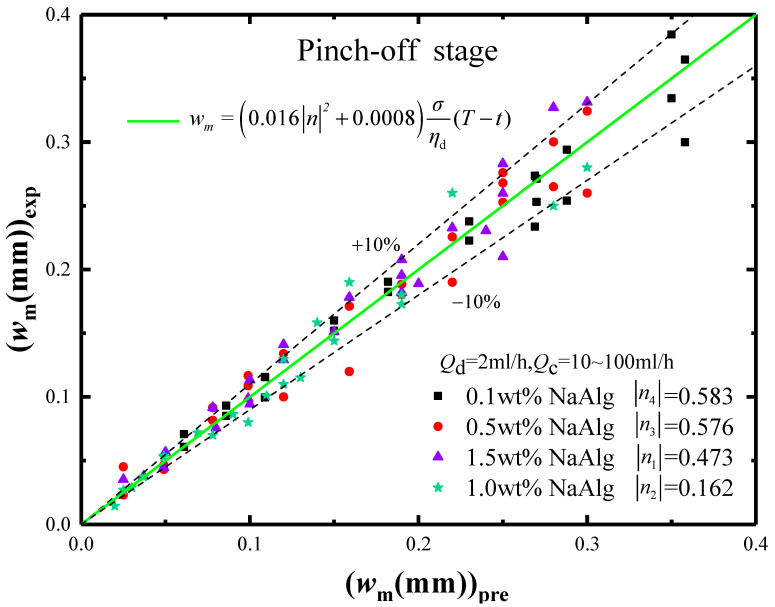
The accuracy analysis of *w_m_* prediction model in the pinch-off stage.

**Figure 13 materials-15-04392-f013:**
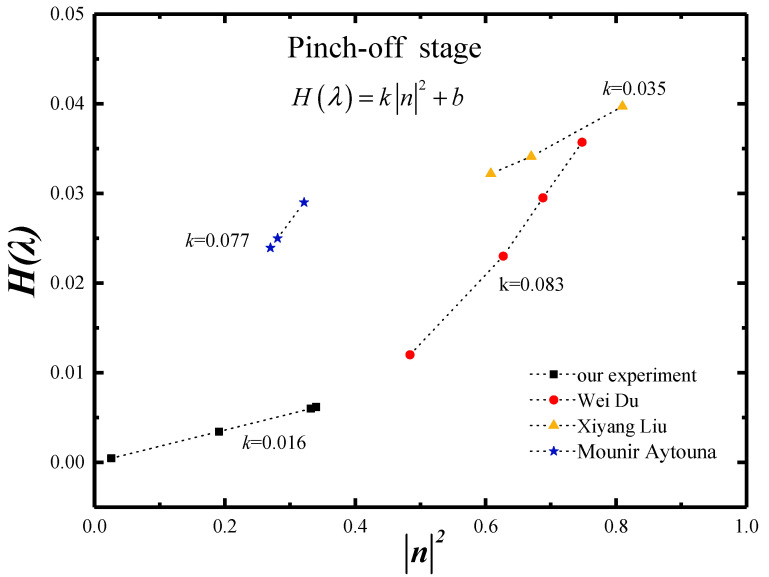
The normalized *H(λ)* function in Stokes scaling law.

**Table 1 materials-15-04392-t001:** The physical properties of two-phase flow (room temperature 25 °C).

Phase	Material	*n*	*ρ* (kg/m^3^)	σ (mN/m)	*η* (Pa·s)		*λ_r_* (s)
					*η*_0_ (Pa·s)	*η_∞_* (Pa·s)	
Dispersed Phase	1.5 wt% NaAlg	0.437	988	46.6	2.61	0.34	0.165
	1.0 wt% NaAlg	0.162	1004	51.3	1.93	0.21	0.467
	0.5 wt% NaAlg	−0.567	998	54.1	1.69	0.07	0.284
	0.1 wt% NaAlg	−0.583	1006	41.4	1.31	0.02	0.272
Continuous Phase	Oil	1	896	--	0.058		--

**Table 2 materials-15-04392-t002:** The normalization model of minimum neck width *w_m_* in the squeezing stage.

	Micro-Droplet	Scaling Law	Authors
Newtonianian fluid	Glycerol solution	$w_{m}=k{\left(T-t\right)}^{\alpha }$	Ma [17] (2021)
	Silicone oil		Fu [39] (2016)
	Glycerol solution		Sun [38] (2018)
Non-Newtonian fluid	PMMA	$w_{m}=k{\left(T-t\right)}^{\alpha }$	Xin [37] (2019)
	CMC solutions		Du [21] (2018)
	NaAlg solution	$w_{m}=k{\left(\left(T-t\right)\cdot \frac{Q_{d}}{Q_{c}}\right)}^{\alpha \left|n\right|}$	This work

**Table 3 materials-15-04392-t003:** The fitting model of *w_m_* with different |*n*| and *Q_d_*/*Q_c_* in the pinch-off stage.

	$Q_{d}/Q_{c}=2/10$	$Q_{d}/Q_{c}=2/20$	$Q_{d}/Q_{c}=2/40$	$Q_{d}/Q_{c}=2/100$
$\left|n_{4}\right|=0.583$	$w_{m}=0.019\left(T-t\right)$	$w_{m}=0.019\left(T-t\right)$	$w_{m}=0.019\left(T-t\right)$	$w_{m}=0.019\left(T-t\right)$
$\left|n_{3}\right|=0.576$	$w_{m}=0.017\left(T-t\right)$	$w_{m}=0.017\left(T-t\right)$	$w_{m}=0.017\left(T-t\right)$	$w_{m}=0.017\left(T-t\right)$
$\left|n_{1}\right|=0.437$	$w_{m}=0.012\left(T-t\right)$	$w_{m}=0.012\left(T-t\right)$	$w_{m}=0.012\left(T-t\right)$	$w_{m}=0.012\left(T-t\right)$
$\left|n_{2}\right|=0.162$	$w_{m}=0.009\left(T-t\right)$	$w_{m}=0.009\left(T-t\right)$	$w_{m}=0.009\left(T-t\right)$	$w_{m}=0.009\left(T-t\right)$

**Table 4 materials-15-04392-t004:** The normalization model of minimum neck width *w_m_* in the pinch-off stage.

Micro-Droplets	Rheological Model	wt%	|*n*|	*H(λ)*	Our Model of *w_m_*			Authors
					Scaling Law	*k* & *b*		
CMC Solution	Power	0.10	0.865	0.0357	$\left\{ \begin{array}{l} w_{m}=H\left(\lambda \right)\frac{\sigma }{\eta_{d}}\left(T-t\right)\\ H\left(\lambda \right)=k{\left|n\right|}^{2}+b \end{array}\right.$		*k* = 0.083 *b* = −0.028	Du [21]
		0.25	0.830	0.0295				
		0.50	0.792	0.0228				
		1.00	0.696	0.0160				
CMC Solution	Power	0.1	0.90	0.0397			*k* = 0.035 *b* = 0.011	Liu [22]
		0.3	0.82	0.0341				
		0.5	0.78	0.0332				
Castor Oil	Herschel-Bulkley	68	0.57	0.029			*k* = 0.077 *b* = 0.004	Aytouna [43]
		74	0.53	0.025				
		80	0.52	0.024				
NaAlg Solution	Bird-Carreau	0.1	0.583	0.0062			*k* = 0.016 *b* = 0.0008	This work
		0.5	0.567	0.0059				
		1.0	0.162	0.0012				
		1.5	0.437	0.0039				

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
