# Peer review of "Accelerating Effects of Flow Behavior Index n on Breakup Dynamics for Droplet Evolution in Non-Newtonian Fluids"

_materials, 2022, doi:10.3390/ma15134392_

Round 1

Reviewer 1 Report

The authors study the droplet formation dynamics for a non-newtonian fluid coflowing in oil applying the Bird-Carreau model to modelize the fluid shear thinning. I am sorry to say that I do not think the article is adequate for publishing in the current form.

I have found the manuscript very hard to read. For that reason, I may have missed some of the points the authors may want to make. I apologize for that.

The authors seem to confuse the flow behavior index in the Ostwald approximation, which models the relation between the shear stress and the shear rate, with one of the parameters of the Bird-Carreau empirical model for the viscosity dependence on the shear rate. This is important, as all their discussion is based on this parameter.

The authors study the dynamics of the squeezing stage in terms of the parameter n, however, when changing the concentration, the viscosity is also changing. The model proposed by the authors does not seem to consider that.

About the methods:

  • Details on how the properties were measured are missing.

The liquid-liquid interfacial tension measurements seem to be too high, considering the typical interfacial tension between water and oils.  The accuracy of the measurement may depend on the density difference measurement

The relaxation time should increase with the polymer concentration.

  • Details on the edge detection procedure used to measure the evolution of the neck diameter are missing.
  • When experiments are compared to simulation, details on how the simulation are performed should be described.

Reviewer 2 Report

This paper studied the micro-droplet evolution with NaAlg non-Newtonian fluid in a vertical and coaxial microchannel. The experimental results look interesting. However, the author should put more details on the model and numerical simulation. For example, what is the mathematical form of Bird-Carreau model? What is the numerical method used to get the numerical simulation results?  These missing details made the paper difficult to understand for non-experts.
